# A Qualitative Assessment of Clinical Practice Guidelines and Patterns for Congenital Cytomegalovirus in the United States

**DOI:** 10.3390/ijns9030037

**Published:** 2023-06-30

**Authors:** Stephanie Kalb, John Diaz-Decaro, Harout Tossonian, Andrew Natenshon, Lori Panther, James Mansi, Laura Gibson

**Affiliations:** 1Moderna, Inc., Cambridge, MA 02139, USA; john.diaz-decaro@modernatx.com (J.D.-D.); harout.tossonian@modernatx.com (H.T.); andrew.natenshon@modernatx.com (A.N.); lori.panther@modernatx.com (L.P.); james.mansi@modernatx.com (J.M.); 2Departments of Medicine and of Pediatrics, UMass Chan Medical School, Worcester, MA 01655, USA; laura.gibson@umassmed.edu

**Keywords:** cytomegalovirus, congenital cytomegalovirus, standard of care, clinical practice guidelines, prenatal CMV screening, neonatal CMV screening, CMV awareness, patient engagement

## Abstract

Cytomegalovirus (CMV) infection during pregnancy may result in long-term health problems for children with congenital CMV (cCMV). Currently, no prevention or treatment interventions are approved by the Food and Drug Administration for a cCMV indication. Healthcare provider and public awareness is low, and formal clinical practice guidelines and local practice patterns vary. A pilot study of eight cCMV experts was performed using qualitative semi-structured interviews to better understand clinical practice guidelines and patterns in the United States. Results from participant interviews highlighted the need for better prenatal diagnostic techniques, broader neonatal screening opportunities, and more robust evidence supporting intervention strategies. Healthcare provider and public partnerships are essential for advancing cCMV guidelines and improving care delivery. Our results provide a preliminary knowledge base and framework for developing a consensus cCMV research agenda to address evidence gaps that limit the revision of clinical practice guidelines. The changes in clinical practice patterns that may arise as a result of further research have the potential to reduce risk during pregnancy and improve care for children with cCMV infection.

## 1. Introduction

Human cytomegalovirus (CMV) is a ubiquitous β-herpesvirus that infects most people worldwide at some point during their lives and establishes lifelong infection [1,2,3]. CMV seroprevalence rates range from 44% to 100% depending on the population studied and is highest in developing countries and among women of reproductive age [1,3,4]. It is estimated that 30% of children in the United States have been infected with CMV by 5 years of age [5]. CMV transmission occurs through direct contact with infectious body fluids, including blood, saliva, urine, tears, semen, cervical secretions, and breast milk. In immunocompetent individuals, CMV infection is generally asymptomatic or associated with mild illness [5]. However, in a pregnant person, vertical transmission of CMV to the fetus can result in congenital CMV (cCMV) infection, which can cause stillbirth or severe health complications [6,7,8,9]. Both primary (i.e., initial infection) and non-primary (i.e., reactivation from latency or reinfection with a new virus) CMV infection during pregnancy are associated with the risk of fetal transmission and severe disease [10,11].

Approximately 10% to 15% of neonates with cCMV infection are born with clinically apparent effects such as jaundice, microcephaly, low birth weight, or sensorineural hearing loss, while the remainder of neonates with CMV infection appear clinically normal at birth [8,12,13]. For approximately 40% to 60% of the symptomatic neonates and 10% to 15% of the asymptomatic neonates, the abnormalities are long term; of all infants born with cCMV infection, nearly 20% will experience permanent neurodevelopmental disabilities [8,12,13]. Although new approaches are being studied [14], no CMV treatments are approved for use in pregnancy, and no vaccine is available [15]. Behavioral strategies to minimize contact with saliva or urine from young children can limit the risk of transmission. Awareness of cCMV remains generally low, and perinatal healthcare workers seldom address risk with patients despite the severe and long-term impact that cCMV infection can have on children [16,17,18,19,20,21,22,23].

Although many evidence-based clinical practice guidelines (CPGs) for pre-pregnancy and maternal care have been published, few provide comprehensive recommendations for cCMV prevention. The US-based American College of Obstetricians and Gynecologists (ACOG) and Society for Maternal-Fetal Medicine (SMFM) each have issued CPGs related to cCMV; however, these CPGs were last updated in 2015 and 2016, respectively, and thus do not reflect more recent studies [24,25]. While ACOG and SMFM both acknowledge the benefits of behavioral strategies for reducing cCMV infection, neither organization has published explicit guidelines on educating patients about these measures. In contrast, the Society of Obstetricians and Gynaecologists of Canada (SOGC) has recently published a CPG with the goals of increasing awareness of cCMV risk reduction for obstetric providers and their patients [26]. Similarly, CPGs that focus primarily on the management of infants with cCMV infection offer detailed recommendations but have not been updated since their original publication in 2017 [27,28].

Variability among the cCMV CPGs presents an opportunity to update and align them, particularly for prenatal risk reduction. Moreover, understanding CPG utilization cannot only support progress in clinical practice but also guide public health policy. We therefore conducted a qualitative assessment of cCMV CPGs and practice patterns from the perspectives of healthcare professionals and research leaders with expertise in the field of cCMV infection and disease.

## 2. Materials and Methods

### 2.1. Study Design and Setting

A qualitative semi-structured interview study was performed to establish an understanding of cCMV CPGs and practice patterns in the United States, including primary and non-primary CMV infection in pregnant people, from the perspective of cCMV research and clinical experts. Interviews were conducted by video conference between 17 March and 17 June 2022.

### 2.2. Participants

With the aim of recruiting 8–10 professional society members with experience developing CMV CPGs, 15 professionals with expertise in a cCMV-related healthcare field were identified. A letter of invitation was sent via email to recruit the identified professionals between 1 March and 8 May 2022; the experts who indicated their willingness to participate in the study via a response email were contacted by the study researcher to arrange an interview. The participants represented a balanced mix of professional society membership, with ACOG, SMFM, and the American Academy of Pediatrics (AAP) represented most. Seven participants were from diverse geographic regions of the United States, and one was from Canada.

### 2.3. Procedure and Data Collection

Interviews were conducted by one of the authors (SK), along with at least one other non-participant attendee. Interviews lasted approximately 1 h and were concluded when data saturation was achieved. Information regarding participant training, clinical specialty, research activities, and participation in CPG development was collected.

### 2.4. Interview Questions

A semi-structured interview schedule consisting of 9 open-ended questions was used to obtain information from experts, and related prompts were used to guide the direction of interviews. “Clinical practice guideline(s)” referred to formal recommendations published by a professional society or group of experts. “Clinical practice patterns” or “standards of care” referred to patient management approaches that can vary by provider or region and may not follow CPGs exactly. Interview questions included but were not limited to: (a) Which clinical practice guidelines and organizations inform cCMV care? (b) What would drive change in clinical practice guidelines and patterns for CMV screening and prevention for pregnant people? (c) What is needed to drive changes in guidelines and approaches to care for patients with cCMV infection? The interviewer was permitted to follow and encourage elaboration on comments spontaneously introduced by participants to allow more detailed information from different fields of healthcare.

### 2.5. Interview Analyses

Interview responses were analyzed with NVivo (release 1.6; Lumivero), a qualitative analysis software package that facilitates the identification and classification of themes in interviews [29]. Themes were derived from categorical codes organized in a hierarchical parent–child relationship. Parent codes were high-level themes that could be disaggregated, including awareness of CPGs for cCMV, differences between US and international CPGs, factors underlying current practice patterns, and those that may motivate shifts in CPGs and practice (Appendix A). Child codes were specific themes that could be aggregated and included, for example, reference to specific professional societies when discussing CPGs for cCMV care. High-level and specific themes were assessed for binary presence in each interview and for the total number of references among participants.

## 3. Results

### 3.1. Demographic and Professional Data

A total of 15 experts were contacted, and eight agreed to be interviewed. The interviews were conducted with eight healthcare professionals and research leaders in the United States (*n* = 7) and Canada (*n* = 1). The US participants were from the Midwest, South, West Coast, East Coast, and Mid-Atlantic regions, providing balanced geographic and demographic representation. The following professional societies were represented by participants: AAP, SMFM, and ACOG. The Canadian participant was a member of SOGC and was included to provide a comparison with US guidelines. Seven of the participants were practicing physicians with expertise relevant to cCMV infection, and one was an epidemiologist with extensive research experience in CMV. Half of the participants (*n* = 4) specialized in pediatrics, and the other half specialized in obstetrics and gynecology (OB/GYN; *n* = 2) and maternal-fetal medicine (MFM; *n* = 2).

### 3.2. Perceptions of Healthcare Professionals

#### 3.2.1. Current Clinical Practice Guidelines and Patterns for Pregnant Individuals and Neonates

During the interviews, all participants indicated their awareness of existing CPGs for cCMV infection. Guidelines issued by ACOG were referenced by most participants (*n* = 7), and the guidelines from AAP and SOGC were each referenced by half of the participants (*n* = 4; Figure 1). Preferences for CPGs issued by specific societies differed according to clinical subspecialty. Participants suggested that OB/GYNs generally favor ACOG as a source of clinical guidance, whereas MFM and reproductive specialists also consult SMFM or the American Society for Reproductive Medicine (ASRM) CPGs. The AAP was identified as the major professional society guiding care for children with cCMV infection.

Differences in CPGs and practice patterns between the United States and other countries were discussed by the majority of participants (*n* = 5), most frequently the SOGC (Appendix A). According to participants, CPGs and practice in the United States are more conservative and change more slowly than in many other countries. In particular, they contrasted the United States with Israel (*n* = 5), France (*n* = 3), Australia (*n* = 2), New Zealand (*n* = 1), and Italy (*n* = 1), regarding their distinct screening guidelines and research priorities.

Professional opinions about and approaches to screening for primary CMV or cCMV infection were noted as key differences between the United States and other countries. Participants suggested that attitudes toward CMV screening in pregnancy differed substantially between the United States and Canada, noting that the recently updated SOGC CPG was “more open to screening” compared with the United States, where screening is not recommended. With respect to neonatal cCMV screening, practices were noted to vary across the United States from universal to targeted screening. Some participants suggested that the reason for absence of CPGs addressing newborn CMV screening is the perceived lack of benefit or cost-effectiveness without an intervention approved by the Food and Drug Administration (FDA) to offer children with infection. Three participants (two of whom were MFM specialists) expressed the perception that universal newborn cCMV screening is not cost-effective (Appendix A), which highlights a gap in economic research or in awareness and understanding of existing research. While many pediatricians in the United States acknowledge the benefits of universal cCMV screening for improving outcomes, few legislative policies have been introduced beyond targeted screening based on newborn hearing screen results [30,31,32].

#### 3.2.2. Factors Contributing to Inconsistent Prenatal Clinical Practice Guidelines and Patterns

Participants identified several factors that contribute to variation in prenatal CPGs and clinical practice patterns. Access to pregnancy termination and limits on maximum gestational age were cited by two participants as a key determinant of attitudes toward CMV screening in pregnancy in different countries. They noted that without access to late-stage termination, especially in the case of severe fetal abnormalities, individuals may face greater pressure to make a decision about their pregnancy early after CMV diagnosis before fetal prognosis can be determined. Moreover, variable pregnancy termination laws [33,34] may be a reason for healthcare providers’ hesitancy to screen pregnant individuals for CMV, particularly in countries where access to these health services is limited. Five participants focused on gaps in technology and data, including insufficient clinical trial evidence, unreliable diagnostic options, and lack of FDA-approved interventions for CMV infection during pregnancy (Figure 2). However, gaps in available solutions and in evidence to support change in existing practices were also identified as important opportunities that, if addressed through targeted research, could aid in the evolution of formal guidelines and therefore clinical practice. Other factors included lack of relevance for obstetric care (*n* = 3) and lower prioritization of cCMV compared with other concerns during pregnancy (*n* = 2). Neither of the two OB/GYNs interviewed indicated that cCMV was irrelevant or a lower priority in their own clinical practices compared with other perinatal diseases and concerns (Appendix A).

#### 3.2.3. Mediators of Change in Clinical Practice Guidelines and Patterns

CPGs clarify and facilitate the delivery of high-quality care. Overall, six of the eight participants indicated that the release of new or revised CPGs or expert consensus was a strong motivator of change in care delivery. On the other hand, shifts in clinical practice patterns can drive revisions in professional society positions on those practices. For example, participants noted that current trends in legislation supporting newborn cCMV screening in many US states has, by definition, amplified awareness among patients and healthcare providers. In turn, increased awareness encourages parent queries and drives more providers to perform cCMV screening despite absent or limited CPGs. Participants similarly emphasized the documented benefits of early interventions on the long-term outcomes of hearing loss, developmental delay, and other conditions with variable causes, including cCMV infection [35,36]. Evidence of these benefits regardless of etiology provides rationale for cCMV screening and may transform formal CPGs and/or local clinical practice patterns.

A key theme among the seven US study participants was that existing approaches for CMV screening in pregnancy are not reliable or predictive and do not seem cost-effective in the absence of FDA-approved interventions. Some participants (*n* = 4) noted that more robust diagnostic approaches that produce fewer false positives would encourage providers and likely professional organizations to make CMV screening standard practice for pregnant people and neonates. Consistent with this perspective, routine CMV screening in pregnancy is not recommended in CPGs from US societies or international consensus groups, which also cite a lack of appropriate screening methods or interventions [24,25,26,27,37]. Participants highlighted the unmet need for these management resources, especially since most women, whether pregnant or not, believe that maternal and neonatal screening should be offered, and many would opt for CMV screening for themselves during pregnancy [18,38]. Most participants (*n* = 7) indicated that evidence demonstrating an effective intervention to reduce fetal CMV transmission would be the major factor prompting changes in both CPGs and clinical practice patterns (Table 1). The inclusion of routine HIV and hepatitis B screening during pregnancy in response to evidence of effective measures to reduce transmission was used as an analog for the importance of clinical studies in shaping CPGs.

#### 3.2.4. Collaborative Efforts for Improving Clinical Practice Guidelines and Patterns

Participants noted that partnerships between professional organizations, healthcare providers, and patient advocacy groups are essential for progress in the care of children with cCMV infection. Recent changes in screening guidelines for Zika virus during pregnancy [39,40] were cited as an example of how collaboration among all stakeholders can mediate CPG changes despite ambiguity in testing options. Participants identified key areas of focus for these efforts, such as immediately increasing awareness of protocol options for neonatal cCMV screening and generating data in the longer term to reassess the burden of CMV disease and to support progress in CMV management (Table 1). Almost all participants (*n* = 7) indicated that raising both public and professional awareness is particularly crucial for advancing cCMV care in pregnancy and childhood. The Centers for Disease Control and Prevention (CDC), ACOG, and the American Board of Obstetrics and Gynecology (ABOG) were identified as influential bodies that could improve awareness and thus serve as vital stakeholder partners (Appendix A). For example, participants suggested that ABOG could mandate the inclusion of CMV-specific content in continuing education for healthcare providers and that the CDC could shift public and professional opinions of cCMV management with awareness campaigns as it did for hepatitis C in pregnancy.

#### 3.2.5. The Importance of Patient and Family Engagement

There was general consensus among participants (*n* = 7) that patient advocacy groups are essential partners in the effort to advance cCMV care. Experts highlighted that professional societies have begun to include patient voice in their decision-making process and that patients are generally accepting of preventative measures and willing to implement them to reduce risk (Appendix A). Participants provided concrete suggestions about messaging and broad accessibility that collaborative teams could leverage, such as using communication channels that are applicable to the group of interest (e.g., social media for younger individuals), expanding awareness beyond the realm of obstetric and pediatric care to other healthcare providers, and intentionally disseminating information to demographic groups that may have lower awareness due to disadvantage or discrimination.

## 4. Discussion

The current lack of comprehensive and consistent CPGs to inform clinical practice patterns hinders optimal cCMV management and outcomes. From a public health perspective, it should be a priority to assess cCMV gaps in knowledge and limitations in care for pregnant people and neonates. We therefore performed a pilot study to evaluate cCMV CPGs and practices from the perspective of clinical and research specialists in pediatrics and general OB/GYN or MFM with expertise in the field of cCMV infection and disease. While their responses to interview questions varied, a clear pattern of strong views emerged: CPGs lack up-to-date analyses of evidence, and recommendations should therefore be revised. Our study identified this and other commonalities in the interview responses, which created a preliminary foundation of knowledge and an overall framework to inform continued research and progress in the field of cCMV infection and disease.

Newborn screening for cCMV is becoming increasingly common; several US states (Illinois, Iowa, Connecticut, New York, Utah, and Virginia) have mandated targeted cCMV screening based on failed hearing evaluation, and one state (Minnesota) has legislated universal newborn screening for cCMV [30,32]. Although implementation of these screening mandates suggests that change may occur in the absence of strong CPG recommendations, participants identified that clinical practice is predominantly informed by disease-specific guidelines. Interestingly, participants did not refer to valacyclovir in the context of maternal CMV screening, despite emerging clinical data suggesting a potential benefit of valacyclovir for treatment of primary CMV infection during early pregnancy [14]. CPGs that comprehensively address cCMV management will be key in improving care for pregnant people and neonates. As indicated by participants in this study, while the timing of the CPG revision process varies substantially between organizations, novel evidence is the shared motivator that underpins CPG evolution, irrespective of the issuing organization.

Although the study size was small and participant responses were intrinsically anecdotal, the semi-structured nature of interviews focused their attention on areas that were highly relevant to patient care. Furthermore, the study group had broad representation across the United States, including the Midwest, South, West Coast, East Coast, and Mid-Atlantic regions, as well as in Canada. In addition to the small study group and nature of the data, a limitation of this study was inherent bias from the generally similar perspectives among participants towards current CPGs. Furthermore, the inclusion of participants who were largely affiliated with academic hospitals may limit generalization of their views to other practice settings. The results from this pilot study call for the exploration of opinions among healthcare providers from other specialties serving various populations and across global regions. We consider the next steps to be identifying knowledge gaps through a systematic review of the published literature and building a consensus around a core cCMV research agenda based on key gaps. This process should be transparent and inclusive, with the involvement of healthcare providers, researchers, industry representatives, affected patients and families, and other cCMV stakeholders. Targeting a broad audience, including funders, this effort should aim to clarify and prioritize the many important areas of cCMV clinical practice and research, ranging from awareness and public health to screening, antiviral therapies, and vaccine development. All stakeholders affected and/or motivated by cCMV share this common vision in which optimal care will ensure that children are born without a potentially devastating CMV congenital infection.

## Figures and Tables

**Figure 1 IJNS-09-00037-f001:**
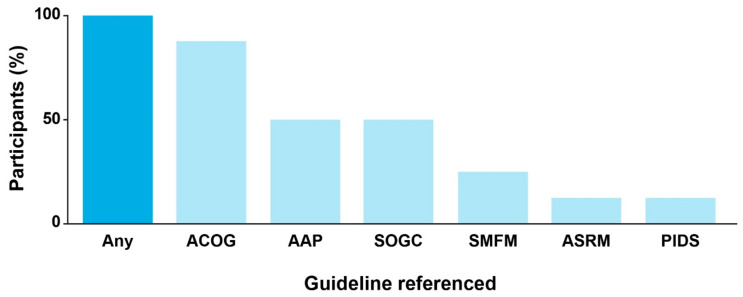
Participant awareness of and reference to clinical practice guidelines for cCMV. The percentage of the eight participants referring to clinical practice guidelines was stratified by the issuing professional body. AAP, American Academy of Pediatrics; ACOG, American College of Obstetricians and Gynecologists; ASRM, American Society for Reproductive Medicine; cCMV, congenital cytomegalovirus; CMV, cytomegalovirus; PIDS, Pediatric Infectious Diseases Society; SMFM, Society for Maternal-Fetal Medicine; SOGC, Society of Obstetricians and Gynaecologists of Canada.

**Figure 2 IJNS-09-00037-f002:**
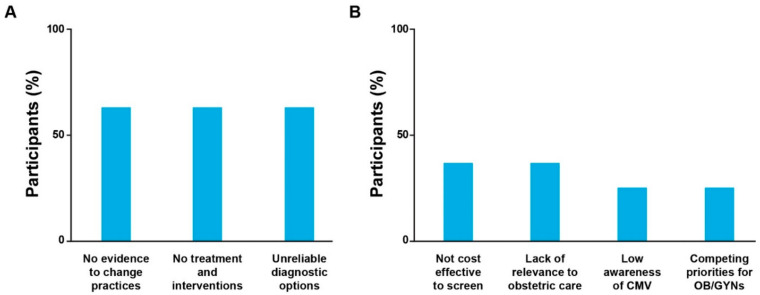
Factors underlying inconsistent cCMV clinical practice guidelines and patterns. The percentage of eight participants citing (**A**) a lack of data and technology and (**B**) environmental conditions or other limitations as factors is shown. cCMV, congenital cytomegalovirus; CMV, cytomegalovirus; OB/GYN, obstetrician/gynecologist.

**Table 1 IJNS-09-00037-t001:** Factors identified as mediators of change in cCMV clinical practice guidelines and patterns, including those noted for collaborative efforts, ranked by the number of eight participants citing each factor.

Theme (Code)	Description of Code	*n* (%)
Factors that would mediate change	Participant referred to factors that would mediate change in cCMV practice patterns	7 (87.5%)
Evidence for treatments	Participant referred to evidence of proven treatments for cCMV disease	7 (87.5%)
Raise awareness	Participant referred to raising awareness of cCMV infection	7 (87.5%)
New or revised guideline publication	Participant referred to the impact of new or revised professional society guidelines on clinical practice	6 (75%)
Available vaccine	Participant referred to the availability of a vaccine for CMV	5 (62.5%)
Improved diagnostic options	Participant referred to a need for improved diagnostic and screening options	4 (50%)
Evidence for the benefits of prevention or treatment	Participant referred to a need for evidence generation regarding preventative measures and treatments	3 (37.5%)
Achievable near-term efforts	Participant referred to immediate actions that may mediate change	2 (25%)

Total number of participants = 8; *n*, number of participants referencing the specific code of interest. Specific themes are shown in white rows below the parent theme in gray. cCMV, congenital cytomegalovirus; CMV, cytomegalovirus.

## Data Availability

The authors declare that the data supporting the findings of this study are available with this article and Appendix A. Excerpts from interview transcripts were adapted and anonymized to ensure participant confidentiality.

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
