# Peer review of "A Qualitative Assessment of Clinical Practice Guidelines and Patterns for Congenital Cytomegalovirus in the United States"

_2409-515X, 2023, doi:10.3390/ijns9030037_

Round 1
Reviewer 1 Report
Congratulations on this quantitative study exploring the understanding of cCMV CPGs and practice patterns among health care professionals in the US. What this study adds is what we, as health professional who practise in this area are aware of, or encounter in our practice, elucidated in writing in a study.
I found the study informative. The Supplemental material and the responses provided interesting and useful background as to how different specialities think about the issues (so, sub-speciality focus). There were also interesting areas of misconception among some health care professionals re: “no FDA approved intervention” was seen as a reason for not advocating “screening”, without considering that ‘knowledge” e.g. establishing a timely diagnosis of congenital CMV is empowering and facilitates discussion, minimises an odyssey of investigations, facilitates appropriate counselling and follow up.
Some comments
1. Methods: this needs more clarity. There may have been biased selection of participants. Why were only 15 experts contacted? Were they all from the same area of the US? Are your participants representative of the US (as your title implies). How were these participants selected?
2. Were all these participants medical, and practicing physicians? I think this is implied but please state this. There is a difference for example if they were all researchers with no practical involvement or nursing/midwifery etc
3. There was a 50% response for participation. The number of participants was small (n = 8). Any reason for this? The US is big and so many health care facilities involved in the care of women with CMV and the infants. Eight participants in this big field seems very small and thus I wonder about the applicability of your study findings across the US
4. It would have been interesting to have also include the focus/consumer groups in this survey
5. I note that the primary author is associated with Moderna, one of the forerunners in the CMV vaccine arena (mRNA CMV vaccine currently in trials). I note the Conflict of Interest statements however appropriately declares this
6. I may be misinterpreting the intent, but please note that “universal screening” CMV screening in pregnancy or for congenital CMV in newborns is also not standard of practice in the Australian (or NZ) CPGs which is outlined in ref 27 (this CPG has US representation)
7. For a reference into the current US CMV in pregnancy/CMV policies : suggest to include/cite the excellent article by Yassine BB, Hulkower R, Dollard S, Cahill E, Lanzieri T. A Legal Mapping Assessment of Cytomegalovirus-Related Laws in the United States. Journal of public health management and practice : JPHMP 2022; 28(2): E624-e9.
You have cited Pes MH, 2022 which is a excellent article as well but Yassine provides a more comprehensive profile
8. There is a potential, emerging intervention in early primary CMV infection in pregnancy (valaciclovir) as per the study by Shahar-Nissan K, Pardo J, Peled O, et al. Valaciclovir to prevent vertical transmission of cytomegalovirus after maternal primary infection during pregnancy: a randomised, double-blind, placebo-controlled trial. Lancet (London, England) 2020; 396(10253): 779-85 that may be compelling as a reason for universal screening in pregancy. It is interesting this was not referred by your participants (as far as I can see). It could have been raised in your Discussions
9. Your Discussion should include the limitations of your study/survey. The small number of participants is one. We also need to know more about who your participants were. I am uncomfortable about this aspect of your study as it is unclear how representative your participants are of the US practice or understanding, nor if you have a biased set of participants.
10. Perhaps your title may need to be revised to " a subset ot USA" if the participants are from only one region /area in the US
Author Response
Reviewer 1
Congratulations on this quantitative study exploring the understanding of cCMV CPGs and practice patterns among health care professionals in the US. What this study adds is what we, as health professional who practise in this area are aware of, or encounter in our practice, elucidated in writing in a study. I found the study informative. The Supplemental material and the responses provided interesting and useful background as to how different specialties think about the issues (so, sub-specialty focus). There were also interesting areas of misconception among some health care professionals re: “no FDA approved intervention” was seen as a reason for not advocating “screening”, without considering that ‘knowledge” e.g. establishing a timely diagnosis of congenital CMV is empowering and facilitates discussion, minimises an odyssey of investigations, facilitates appropriate counselling and follow up.
Response: We thank the reviewer for their kind feedback and valuable insights to improve this manuscript.
Specific comments:
- Methods: this needs more clarity. There may have been biased selection of participants. Why were only 15 experts contacted? Were they all from the same area of the US? Are your participants representative of the US (as your title implies). How were these participants selected?
Response: The goal of the study was to investigate priorities and concerns around advancing CMV and cCMV standards of care in the United States by interviewing 8-10 experts who were professional society members with experience enacting CMV clinical practice guidelines (CPGs). Initially, 15 experts were contacted as we anticipated this would provide the planned number of 8-10 participants for the interviews. Ultimately, this approach yielded a balanced mix of professional society members who also had experience enacting CMV CPGs. The participants represented several different professional societies; the 7 US participants were from various regions in the United States, providing balanced demographic and geographic representation. One Canadian participant, a member the Society of Obstetricians and Gynecologists of Canada (SOGC), was also included to provide a comparison with US guidelines. The manuscript text has been revised to reflect these details.
Revised text (Materials and Methods, page 5): With the aim of recruiting 8-10 professional society members with experience enacting CMV CPGs, 15 professionals with expertise in a cCMV-related healthcare field were identified. A letter of invitation was sent via email to recruit the identified professionals between March 1 and May 8, 2022; the experts who indicated their willingness to participate in the study via a response email were contacted by the study researcher to arrange an interview. The participants represented a balanced mix of professional society membership with ACOG, SMFM, and American Academy of Pediatrics (AAP) represented most. Seven participants were from diverse geographic regions of the United States, and 1 was from Canada.
Revised text (Results, page 7): “The interviews were conducted with 8 healthcare professionals and research leaders in the United States (n = 7) and Canada (n = 1). The US participants were from the Midwest, South, West Coast, East Coast, and Mid-Atlantic regions, providing balanced geographic and demographic representation. The following professional societies were represented by participants: AAP, SMFM, and ACOG. The Canadian participant was a member of SOGC and was included to provide a comparison with US guidelines.”
- Were all these participants medical, and practicing physicians? I think this is implied but please state this. There is a difference for example if they were all researchers with no practical involvement or nursing/midwifery etc
Response: Seven of the participants were practicing medical physicians and 1 was an epidemiologist with CMV research experience. This point has been clarified in the revised manuscript.
Revised text (Results, page 7): “Seven of the participants were practicing physicians with expertise relevant to cCMV infection and 1 was an epidemiologist with extensive research experience in CMV.”
- There was a 50% response for participation. The number of participants was small (n = 8). Any reason for this? The US is big and so many health care facilities involved in the care of women with CMV and the infants. Eight participants in this big field seems very small and thus I wonder about the applicability of your study findings across the US
Response: Interviews were conducted during early 2022, and although it is unclear why the response rate was low, this may have been associated with a combination of factors, including an increase in clinical duties during the pandemic and video meeting fatigue. Additionally, as noted in response to Comment 1, despite the relatively small number of participants, they represented diverse geographic regions of the United States, thereby broadened the applicability of the findings.
- It would have been interesting to have also include the focus/consumer groups in this survey
Response: While we agree with the observation that other groups may have provided interesting insights, we believe the inclusion of focus/consumer groups is out of the scope for the current study.
- note that the primary author is associated with Moderna, one of the forerunners in the CMV vaccine arena (mRNA CMV vaccine currently in trials). I note the Conflict of Interest statements however appropriately declares this
Response: We thank the reviewer for noting this.
- I may be misinterpreting the intent, but please note that “universal screening” CMV screening in pregnancy or for congenital CMV in newborns is also not standard of practice in the Australian (or NZ) CPGs which is outlined in ref 27 (this CPG has US representation)
Response: The Discussion has been revised to include text stating that routine CMV screening is not recommended by international guidelines and organizations.
Revised text (Discussion, page 11): “Consistent with this perspective, routine CMV screening in pregnancy is not recommended in CPGs from US societies or international consensus groups, which also cite a lack of appropriate screening methods or interventions.24-27,37”
- For a reference into the current US CMV in pregnancy/CMV policies : suggest to include/cite the excellent article by Yassine BB, Hulkower R, Dollard S, Cahill E, Lanzieri T. A Legal Mapping Assessment of Cytomegalovirus-Related Laws in the United States. Journal of public health management and practice : JPHMP 2022; 28(2): E624-e9.
You have cited Pes MH, 2022 which is a excellent article as well but Yassine provides a more comprehensive profile
Response: We agree with the Reviewer that the article by Yassine et al is an important source of information regarding the current US policies for CMV in pregnancy. Therefore, this publication has been added (reference 32) to the revised manuscript on pages 9 and 15.
- There is a potential, emerging intervention in early primary CMV infection in pregnancy (valaciclovir) as per the study by Shahar-Nissan K, Pardo J, Peled O, et al. Valaciclovir to prevent vertical transmission of cytomegalovirus after maternal primary infection during pregnancy: a randomised, double-blind, placebo-controlled trial. Lancet (London, England) 2020; 396(10253): 779-85 that may be compelling as a reason for universal screening in pregnancy. It is interesting this was not referred by your participants (as far as I can see). It could have been raised in your Discussions
Response: We agree that this is an interesting observation and as such, have noted in the revised manuscript the lack of reference by participants to valaciclovir in the context of CMV screening during pregnancy.
Revised text (Discussion, page 15): “Interestingly, participants did not refer to valacyclovir in the context of maternal CMV screening, despite emerging clinical data suggesting a potential benefit of valacyclovir for treatment of primary CMV infection during early pregnancy.14”
- Your Discussion should include the limitations of your study/survey. The small number of participants is one. We also need to know more about who your participants were. I am uncomfortable about this aspect of your study as it is unclear how representative your participants are of the US practice or understanding, nor if you have a biased set of participants
Response: The US participants were from diverse regions and this point has been clarified in the revised manuscript, as described in response to Comment 1. Additionally, the Discussion has been expanded to include a more detailed description of study limitations, including the intrinsic biases introduced by the affiliation of participants with academic hospitals that predominantly provide care for urban populations and the skewed representation of opinions in favor of CPG revisions.
Revised text (Discussion, page 15): “Although the study size was small and participant responses were intrinsically anecdotal, the semi-structured nature of interviews focused their attention on areas that were highly relevant to patient care. Furthermore, the study group had broad representation across the United States, including the Midwest, South, West Coast, East Coast, and Mid-Atlantic regions, as well as in Canada. In addition to the small study group and nature of the data, a limitation of this study was inherent bias from the generally similar perspectives among participants towards current CPGs. Furthermore, the inclusion of participants who were largely affiliated with academic hospitals may limit generalization of their views to other practice settings. The results from this pilot study call for the exploration of opinions among healthcare providers from other specialties serving various populations, and across global regions.”
- Perhaps your title may need to be revised to " a subset ot USA" if the participants are from only one region /area in the US
Response: The participants were from a wide range of geographic regions in the United States, including the Midwest, South, West Coast, East Coast, and Mid-Atlantic regions.
.
Reviewer 2 Report
Dear Authors
This is a very important topic, and I fully agree with the call for greater public awareness and filling of knowledge gaps.
These objectives might have been better fulfilled by using a narrative review, solicited from an expert in the field by the journal directly.
Nevertheless, here are my comments on the presented manuscript:
The sampling method, being purposive - is inherently a subjective one. With this being a qualitative study design, greater transparency is required on how the sample was chosen. More especially as the authors are all associated with a commercial entity, with interests in this field. The same authors then conducted the interviews, and analysed the data, with no mention of measures to reduce bias in their methods. This would create questions around Conflict of Interest in the reader's mind, and adversely impact on their impression of the study's trustworthiness. In the discussion, the "varied sampling" is presented as a strength of the study.
The readers of this journal, who I would guess are predominantly readers of quantitative research, would likely value a better description of how this was achieved, in the section entitled 2.2 Participants, on page 2. Purposive sampling can be stratified hierarchically, or possibly suitable for this study – can be in “Cells” – example, experts from: Public Health, Obstetrics, Paediatrics, Behavioural Medicine, Vaccinology, Virology, Family medicine, Policy makers. A further categorization might include those who have published research on: cCMA prevention, screening for cCMV, cCMV treatment. A truly varied sampling technique would strive to include these various categories of experts, and assure the reader of the scientific rigour.
Perhaps the conclusion might not just call for filling of gaps in knowledge, but also a systematic review of the evidence, to ensure that we know what those gaps are. The authors should also place their recommendations in the framework of public health strategic objectives e.g. CDC in order to better emphasize the need for policy changes.
Some specific comments:
Abstract, line 2: - there are cCMV treatment interventions approved by the FDA e.g. Ganciclovir. Did you mean, there are no approved no Prevention or screening interventions?
Figure 2 – figure label – “Factors underlying cCMV…” should perhaps be “Factors underlying inconsistent cCMV…”
page 6 – “6 of the 8 participants” – there is a typo – delete the letter “t”
page 7 “ … screening methods or interventions” – References 36 and 37 are not from the US.
References – References 26 and 37 are duplicates.
Author Response
This is a very important topic, and I fully agree with the call for greater public awareness and filling of knowledge gaps.
These objectives might have been better fulfilled by using a narrative review, solicited from an expert in the field by the journal directly.
Nevertheless, here are my comments on the presented manuscript:
Response: We thank the reviewer for their valuable feedback to improve this manuscript. Point-by-point responses to each comment are included below.
Specific comments:
- The sampling method, being purposive - is inherently a subjective one. With this being a qualitative study design, greater transparency is required on how the sample was chosen. More especially as the authors are all associated with a commercial entity, with interests in this field. The same authors then conducted the interviews, and analysed the data, with no mention of measures to reduce bias in their methods. This would create questions around Conflict of Interest in the reader's mind, and adversely impact on their impression of the study's trustworthiness. In the discussion, the "varied sampling" is presented as a strength of the study.
Response: We removed the term "varied sampling" from the list of strengths of the study. We agree that that this term could be misleading.
- The readers of this journal, who I would guess are predominantly readers of quantitative research, would likely value a better description of how this was achieved, in the section entitled 2.2 Participants, on page 2. Purposive sampling can be stratified hierarchically, or possibly suitable for this study – can be in “Cells” – example, experts from: Public Health, Obstetrics, Paediatrics, Behavioural Medicine, Vaccinology, Virology, Family medicine, Policy makers. A further categorization might include those who have published research on: cCMA prevention, screening for cCMV, cCMV treatment. A truly varied sampling technique would strive to include these various categories of experts, and assure the reader of the scientific rigour.
Response: Similar to our response above regarding the use of “varied sampling”, we agree that the use of “purposive” to describe our sampling may be misleading given that our sampling was not stratified into cells as described above. Therefore, we have removed this term from the Methods section in addition to the removal of the term “varied sampling” to better describe the sample characteristics in the Discussion.
- Perhaps the conclusion might not just call for filling of gaps in knowledge, but also a systematic review of the evidence, to ensure that we know what those gaps are. The authors should also place their recommendations in the framework of public health strategic objectives e.g. CDC in order to better emphasize the need for policy changes.
Response: We agree with the reviewer and do indeed consider a systematic review of evidence an essential step in identifying knowledge gaps. Therefore, the conclusion has been expanded to include a systematic literature review as part of next steps.
Revised text (Discussion, page 15): “We consider next steps to be identifying knowledge gaps through a systematic review of published literature and building consensus around a core cCMV research agenda based on key gaps.”
- Abstract, line 2: - there are cCMV treatment interventions approved by the FDA e.g. Ganciclovir. Did you mean, there are no approved no Prevention or screening interventions?
Response: We thank the reviewer for this comment, but respectfully note that there are no FDA-approved treatments currently indicated for cCMV. Per the prescribing information, ganciclovir is only indicated for treatment of CMV retinitis in immunocompromised adults or prevention of CMV disease in adult transplant recipients at risk for CMV disease, while valganciclovir is approved for pediatric use only for prevention of CMV disease in kidney or heart transplant patients at high risk. As such, we have kept this statement largely as originally written with the following minor revisions for clarity:
Revised text (Abstract, page 2): “Currently, no prevention or treatment interventions are approved by the Food and Drug Administration for a cCMV indication, healthcare provider and public awareness is low, and formal clinical practice guidelines and local practice patterns vary.”
- Figure 2 – figure label – “Factors underlying cCMV…” should perhaps be “Factors underlying inconsistent cCMV…”
Response: The label for Figure 2 has been revised accordingly.
- page 6 – “6 of the 8 participants” – there is a typo – delete the letter “t”
Response: We thank the reviewer for bringing this error to attention and note that this has been corrected in the revised manuscript.
- page 7 “ … screening methods or interventions” – References 36 and 37 are not from the US.
Response: As described in comment 6 from Reviewer 1, text in the Discussion has been revised to include that routine CMV screening is not recommended by international organizations. References 26 (formerly reference 37) and 36 are now cited appropriately to describe international CPGs.
Revised text (Discussion, page 11): “Consistent with this perspective, routine CMV screening in pregnancy is not recommended in CPGs from US societies or international consensus groups, which also cite a lack of appropriate screening methods or interventions.24-27,37”
- References – References 26 and 37 are duplicates
Response: The duplicate reference to Boucoiran et al, 2021 has been removed.
Round 2
Reviewer 2 Report
The comments have been addressed.